

# Derivation of the entropic formula for the statistical mechanics of space plasmas

George Livadiotis

Southwest Research Institute, Space Science & Engineering, San Antonio, TX, USA

5  *Correspondence to*: George Livadiotis (glivadiotis@swri.edu)

**Abstract.** Kappa distributions describe velocities and energies of plasma populations in space plasmas. The statistical origin of these distributions is the non-extensive statistical mechanics. Indeed, the kappa distribution is derived by maximizing the q-entropy of Tsallis under the constraints of canonical ensemble. However, there remains the question what is the physical origin of this entropic formulation. This paper shows that the q-entropy can be derived by adapting the additivity of energy 10 and entropy.

## 1 Introduction

Space plasmas are collisionless and correlated particle systems characterized by a non-Maxwellian behavior, typically described by the formulations of kappa distributions. The origin of this vastly different statistical behavior between classical systems and space plasmas is the manifestation of correlations between the plasma particles.These systems are characterized 15  by long-range interactions that induce correlations resulting to a collective behavior among particles (e.g., see Jund et al. 1995; Salazar & Toral 1999; Villain 2008; Tsallis 2009; Grassi 2010; Tirnakli & Borges 2016). The induction of any type of correlations among particles (more accurately, among particle energies or particle phase-space) departs the system from thermal equilibrium to be re-stabilized to other stationary states out of thermal equilibrium described by kappa distributions.

Kappa distributions describe numerous space plasma populations. Several examples are the following: (i) *inner* 20  *heliosphere*, including solar wind (e.g., Maksimovic et al. 1997; Pierrard et al. 1999; Mann et al. 2002; Marsch 2006; Zouganelis 2008; Štverák et al. 2009; Livadiotis and McComas 2013a; Yoon 2014; Pierrard and Pieters 2015; Pavlos et al. 2016), solar spectra (e.g., Dzifčáková and Dudík 2013; Dzifčáková et al. 2015), solar corona (e.g., Owocki and Scudder 1983; Vocks et al. 2008; Lee et al. 2013; Cranmer 2014), solar energetic particles (e.g., Xiao et al. 2008; Laming et al. 2013), corotating interaction regions (e.g., Chotoo et al. 2000), and solar flares related (e.g., Mann et al. 2009; Livadiotis and 25  McComas 2013b; Bian et al. 2014; Jeffrey et al. 2016); (ii) *planetary magnetospheres*, including magnetosheath (e.g., Formisano et al. 1973; Ogasawara et al. 2013), magnetopause (e.g., Ogasawara et al. 2015), magnetotail (e.g., Grabbe 2000), ring current (e.g., Pisarenko et al. 2002), plasma sheet (e.g., Christon 1987; Wang et al. 2003; Kletzing et al. 2003), magnetospheric substorms (e.g., Hapgood et al. 2011), Aurora (e.g., Ogasawara et al. 2017), magnetospheres of giant planets, such as Jovian (e.g., Collier and Hamilton 1995; Mauk et al. 2004), Saturnian (e.g., Dialynas et al. 2009; Livi et al.



2014; Carbary et al. 2014), Uranian (e.g., Mauk et al. 1987), Neptunian (Krimigis et al. 1989), magnetospheres of planetary moons, such as Io (e.g., Moncuquet et al. 2002) and Enceladus (e.g., Jurac et al. 2002), cometary magnetospheres (e.g., Broiles et al. 2016a; 2016b); (iii) *outer heliosphere and the inner heliosheath* (e.g., Decker and Krimigis 2003; Decker et al. 2005; Heerikhuisen et al. 2008; 2015; Zank et al. 2010; Livadiotis et al. 2011; 2012; 2013; Livadiotis and McComas 2011a;

2012; 2013c; Livadiotis 2014; 2016; Fuselier et al. 2014; Zirnstein and McComas 2015; Zank, 2015); (iv) *beyond the heliosphere*, including HII regions (e.g., Nicholls et al. 2012), planetary nebula (e.g., Nicholls et al. 2013; Zhang et al. 2014), and supernova magnetospheres (e.g., Raymond et al. 2010); and in cosmological scales (e.g., Hou et al. 2017); (iv) *other space plasma-related analyses* (e.g., Milovanov and Zelenyi 2000; Saito et al. 2000; Du 2004; Yoon et al. 2006; 2012; Raadu and Shafiq 2007; Livadiotis 2009; 2015a; 2015c; Tribeche et al. 2009; Hellberg et al. 2009; Livadiotis and McComas

2010b; 2014; Baluku et al. 2010; Le Roux et al. 2010; Eslami et al. 2011; Kourakis et al. 2012; Randol and Christian 2014; 2016; Varotsos et al. 2014; Fisk and Gloeckler 2014; Viñas et al. 2014, 2015; Ourabah et al. 2015; Dos Santos et al., 2016; Nicolaou and Livadiotis, 2016). (See also the book: Livadiotis 2017a, and references therein.)

Empirical kappa distributions have been introduced in mid-60's by Binsack (1966), Olbert (1968), and Vasyliũnas (1968), while their connection with statistical mechanics was shown and studied in detail in about half century later (see

Livadiotis and McComas 2009, and references therein). In particular, the statistical origin of these distributions is now widely accepted to be determined within the framework of non-extensive statistical mechanics (Tsallis 2009). This is a consistent generalization of the classical statistical mechanics, which is based on a mono-parametric ($q$-index) entropic formula (Tsallis 1988). The theoretical $q$-exponential distribution, which results from the maximization of entropy in the canonical ensemble, has the same formulation with the empirical kappa distribution; the two distributions are identical under

the transformation of their characteristic indices ($q = 1 + 1/\kappa$).

Having attained a consistent connection of the mathematical model of kappa distributions with the physical means of entropy maximization does not precisely answer the main question regarding the origin of these distributions. We have shifted the modeling from the distributions to the entropic formulation. Therefore, we may understand now that the statistical origin of kappa distributions is given by the Tsallis entropy maximization in the canonical ensemble, but still, the origin of

this specific entropic formulation remains unknown.

Certainly, there are various mechanisms responsible for generating kappa distributions in space and other plasmas; for example, the presence of pickup ions (Livadiotis and McComas 2010a; 2011a) or weak turbulence (Yoon et al. 2012; Yoon 2014). Moreover, kappa distributions belong to the framework of non-extensive statistical mechanics. Thus, once a kappa distribution is generated and stabilized in a plasma population, the whole "tool package" of non-extensive statistical

mechanics is applicable for describing the statistical physics of this population; for instance, the entropy is given by the Tsallis formulation, while the temperature can be determined by the mean kinetic energy.

Here, we do not argue about whichever mechanisms generate kappa distributions in space plasmas, but for the physical reasons that these distributions sustain themselves in space plasmas once generated. The typical answer is that this is an effect of the presence and preservation of correlations in the collisionless environment that governs space plasmas. The





collisionless environment is preserving the energy leading to the additivity of energy: The energy of a multi-particle state is the sum of the energies of all the involved one-particle states. On the other hand, the preservation of local correlations among particles creates a conceptual separation of particles in correlation clusters. Debye spheres are correlation clusters that may include up to trillions of particles, since space plasmas are weakly coupled (Bryant 1996; Rubab and Murtaza 2006; Gougam

and Tribeche 2011; Livadiotis and McComas 2014). This structure can lead to the additivity of entropy: The entropy of a multi-particle state is the sum of the entropies of all the involved one-particle states.

The purpose of this paper is to show that there is a deeper connection of Tsallis $q$-entropy and space plasmas: Namely, we will show that two simple first-principles such as the additive energy and additive entropy, which apply to plasma particle populations, are sufficient for indicating the specific formula of $q$-entropy (Figure 1). Therefore, the main

objective of this work is to demonstrate the theory which determines that the entropic form given by the q-entropy formula proposed originally by Tsallis (1988) follows from certain assumptions regarding the (microscopic) state of the system. The importance of this discussion for the (space) plasma physics community resides mostly on the fact that the kappa velocity/energy distribution functions, ubiquitously observed in space and astrophysical environments, can be derived from the maximization of the q-entropy, under the constraints of a canonical ensemble.

In Section 2 we describe the physical motive of this paper in detail. In Section 3 we show in detail a similar property for both the entropic formalisms of BG and Tsallis: The entropy is non-additive in general for some arbitrary probability distribution; but it can become additive specifically for the canonical probability distribution (the one that maximizes the corresponding entropy). In Section 4 we show how we can determine the entropic formula appropriate for describing the plasma particle populations, simply by setting two first-principles properties, obvious for collisionless plasmas: energy and

entropy are additive, at least macroscopically. Finally, Section 5 briefly summarizes the conclusions.

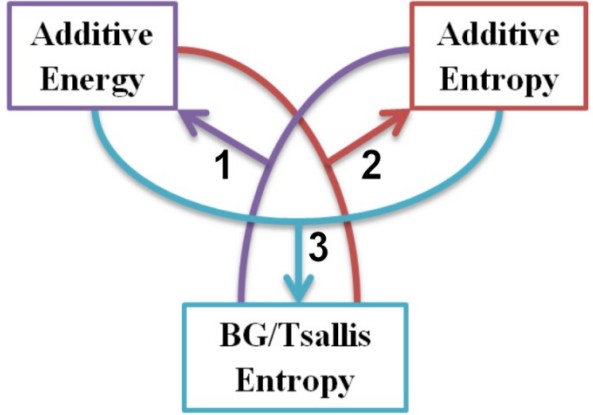

**Figure 1: The infogram indicates the following triplet of concepts: (i) additive energy, (ii) additive entropy, and (iii) BG or Tsallis entropic formulation. Given any two out of the three features, the third can be derived. It is already known that BG or Tsallis**
**entropic formulations can lead to additive entropy if the energy is also additive (red arrow 2). In the same way, it can be shown that these entropic formulations can lead to additive energy if the entropy is additive (purple arrow 1). The objective here is to show that the entropic formula can be derived from the additivity of energy and entropy (blue arrow 3).**



## 2 Physical Motive

Classical Boltzmann Gibbs (BG) statistical mechanics characterizes systems with no correlations among particle velocities or energies. Therefore, the joint two-particle probability distribution can be expressed as the product of the one-particle identical independent discrete distributions; i.e., labelling the two particles with A and B, $p_{ij}^{A+B} = p_i^A \cdot p_j^B$. Hereafter, we

consider a particle system described by a discrete energy spectrum $\{\varepsilon_k\}_{k=1}^W$, which is associated with a discrete probability distribution $\{p_k\}_{k=1}^W$. The same semantics is used when the system is separated in two subsystems A and B, where the two-particle distribution describes a two-particle state, with one particle at each subsystem. The logarithm of the probability is an additive function, $\ln p_{ij}^{A+B} = \ln p_i^A + \ln p_j^B$, from which we obtain the additivity of entropy, $S^{A+B} = S^A + S^B$. For special cases, however, where the independent relation does not apply, $p_{ij}^{A+B} \neq p_i^A \cdot p_j^B$, the entropy is non-additive,

$S^{A+B} \neq S^A + S^B$. The logical reciprocate to the statements above is provided by the uniqueness theorem of Shannon (1948) and Khinchin (1960) that showed that under the assumption of the additivity of entropy (and other basic properties of entropy), the sufficient and necessary entropic form is given by the BG formula.

Non-extensive statistical mechanics characterizes systems with correlations among particles, $p_{ij}^{A+B} \neq p_i^A \cdot p_j^B$. For special systems, however, where the independent relation still applies, the entropy is non-additive, $S^{A+B} \neq S^A + S^B$; in

particular, a square, nonlinear term is added to the summation, $S^{A+B} = S^A + S^B + (1-q)S^A S^B$ for some value of the entropic parameter $q$ (where we set the Boltzmann constant $k_B$ to 1). Note that the logical reciprocates exists also for this case, as shown by dos Santos (1997) and Abe (2000); namely, under the assumption of the mentioned non-additive property (and other basic properties of entropy), the sufficient and necessary entropic form is given by the q-entropic formula of Tsallis (1988).

Another property that is related with the additivity but is even more subtle and difficult to ascertain is the *extensivity* of the entropy. A non-additive entropy may be assumed to be also non-extensive, but it is the inverse assumption that is always correct, i.e., non-extensivity implies non-additivity). Nevertheless, certain correlations, expressed by the relation $p_{ij}^{A+B} = g^{-1}[g(p_i^A) + g(p_j^B)]$ for some function $g$, can make the Tsallis entropy additive, and thus, recover its extensivity (e.g., Tsallis et al. 2005, Ruseckas 2015). In his book, C. Tsallis (2009) goes to great lengths to show that it is possible to

find systems for which the BG entropy is *not* extensive. On the other hand, he argues that there are certain systems for which the entropic form can be extensive, for certain values of the entropic index $q$. In fact, he mentions in the preface that the term "nonextensive entropy" is somewhat incorrect in this sense, but it stuck for historical reasons.

The two statistical formalisms, classical BG and Tsallis non-extensive, have the common property that their entropy becomes additive for some specific function $g$ in the relation $p_{ij}^{A+B} = g^{-1}[g(p_i^A) + g(p_j^B)]$, that is, $g(x) \propto \ln(x)$ and



$g(x) \propto (x^{q-1}-1)/(q-1)$, respectively; (the latter is related to the $q$-deformed logarithm; see: Silva et al. 1998; Yamano 2002).

It is important that the above probability relation is a characteristic feature of the canonical probability distribution in both the formalisms. In other words, the probability distribution that maximizes the BG entropy under the constraints of the

canonical ensemble obeys to correlations expressed by $g(x) \propto \ln(x)$ or $p_{ij}^{A+B} = p_i^A \cdot p_j^B$, which means zero correlation (due to the factorization of the exponentials, Livadiotis and McComas 2011b) that makes the entropy additive. Also, the probability distribution that maximizes the Tsallis entropy under the same constraints obeys to specific correlations expressed by $g(x) \propto (x^{q-1}-1)/(q-1)$ or $(p_{ij}^{A+B})^{q-1} = (p_i^A)^{q-1} + (p_j^B)^{q-1} - 1$, which makes again the entropy additive. In Section 3 we show in detail this similar property of the two statistical formalisms.

Then, we may ask: Is the above described property of BG and Tsallis entropies a general feature of any physically meaningful entropic function? Or, can we reverse the question, and ask which specific entropic function obeys to the above properties? It will be really intriguing if we can determine the entropic formula appropriate for describing the plasma particle populations, simply by setting the following two first-principles properties: (1) additive energy, (2) additive entropy, i.e., the probability distribution derived by maximizing the entropy under the constraints of the canonical ensemble, makes the

entropy additive. This will be the main purpose of this paper and will be examined in Section 4.

## 3 Canonical ensemble distributions with additive energy lead to additive entropy

### 3.1 The Gibbs' path

The Gibbs' path (1902) for the maximization of the entropy $S(p_1, p_2, ..., p_W)$ under the constraints of canonical ensemble, i.e., (i) normalization $1 = \sum_{k=1}^{W} p_k$, and (ii) fixed internal energy $U = \sum_{k=1}^{W} p_k \varepsilon_k$, involves maximizing the functional

$$G(p_1, p_2, ..., p_W) = S(p_1, p_2, ..., p_W) + \lambda_1 \sum_{k=1}^{W} p_k + \lambda_2 \sum_{k=1}^{W} p_k \varepsilon_k \quad . \tag{1}$$

Next, we examine the BG and Tsallis entropic formulations.

### 3.2 BG entropy

First, we start from the classical case of BG entropy

$$S(p_1, p_2, ..., p_W) = -\sum_{k=1}^{W} p_k \ln(p_k) \quad , \tag{2}$$

where we ignored the Boltzmann constant $k_B$ for simplicity. Then, setting $(\partial/\partial p_j)G(p_1, p_2, ..., p_W) = 0$ to





$$G(p_1, p_2, ...., p_W) = -\sum_{k=1}^{W} p_k \ln(p_k) + \lambda_1 \sum_{k=1}^{W} p_k + \lambda_2 \sum_{k=1}^{W} p_k \varepsilon_k \ , \tag{3}$$

we find

$$p_j(\varepsilon_j) = \exp(\lambda_1 - 1) \cdot \exp(\lambda_2 \varepsilon_j) \ . \tag{4}$$

We may write Eq.(4) in a logarithmic form, $\ln p_j = \lambda_2 \varepsilon_j + \lambda_1 - 1$. Then, we separate the particle system in two parts A and

5  B, so that each part is a new subsystem for which Eq.(4) holds:

$$\ln p_i^{\mathrm{A}} = \lambda_2 \varepsilon_i^{\mathrm{A}} + \lambda_1 - 1 \quad \text{and} \quad \ln p_j^{\mathrm{B}} = \lambda_2 \varepsilon_j^{\mathrm{B}} + \lambda_1 - 1 \ . \tag{5}$$

The whole system is characterized by the joint probability, $p_{ij}^{\mathrm{A+B}}$, meaning the probability of a particle in the subsystem A

to reside at the state $i$ and a particle in the subsystem B to reside at the state $j$. This is related with the energy $\varepsilon_{ij}^{\mathrm{A+B}}$ of the

two-particle state,

$$\ln p_{ij}^{\mathrm{A+B}} = \lambda_2 \varepsilon_{ij}^{\mathrm{A+B}} + \lambda_1 - 1 \ . \tag{6}$$

Trivially, the energy of the two-particle state energy $\varepsilon_{ij}^{\mathrm{A+B}}$ equals the summation of the energy of each particle (since no

interparticle force is considered), i.e., system's energy is additive:

$$\varepsilon_{ij}^{\mathrm{A+B}} = \varepsilon_i^{\mathrm{A}} + \varepsilon_j^{\mathrm{B}} \ . \tag{7}$$

Hence, by eliminating energies from Eqs.(5,6), we find

$$\ln p_{ij}^{\mathrm{A+B}} + (\lambda_1 - 1) = \lambda_2 \varepsilon_i^{\mathrm{A}} + (\lambda_1 - 1) + \lambda_2 \varepsilon_j^{\mathrm{B}} + (\lambda_1 - 1) = \ln p_i^{\mathrm{A}} + \ln p_j^{\mathrm{B}} \ , \text{or} \tag{8}$$

$$p_{ij}^{\mathrm{A+B}} = p_i^{\mathrm{A}} \cdot p_j^{\mathrm{B}} \cdot e^{-(\lambda_1 - 1)} \ . \tag{9}$$

At this point we recall that the Lagrange multipliers, $\lambda_1$ and $\lambda_2$, are related with the partition function $Z = e^{-(\lambda_1 - 1)}$

and the inverse temperature $\beta = -\lambda_2$, respectively, and they are not necessarily equal for the two subsystems A and B, or the

whole system A+B. Nevertheless, the logarithm of the partition function or $(\lambda_1 - 1)$ is an extensive parameter, i.e.,

20  $(\lambda_1 - 1)^{\mathrm{A+B}} = (\lambda_1 - 1)^{\mathrm{A}} + (\lambda_1 - 1)^{\mathrm{B}}$, while the temperature is not an extensive parameter and can be considered the same

$\lambda_2^{\mathrm{A+B}} = \lambda_2^{\mathrm{A}} = \lambda_2^{\mathrm{B}}$. Then, instead of Eqs.(8,9), we obtain

$$\ln p_{ij}^{\mathrm{A+B}} = \lambda_2 \varepsilon_{ij}^{\mathrm{A+B}} + (\lambda_1 - 1)^{\mathrm{A+B}} = \lambda_2 \varepsilon_i^{\mathrm{A}} + (\lambda_1 - 1)^{\mathrm{A}} + \lambda_2 \varepsilon_j^{\mathrm{B}} + (\lambda_1 - 1)^{\mathrm{B}} = \ln p_i^{\mathrm{A}} + \ln p_j^{\mathrm{B}} \ , \tag{10}$$

which clearly shows that the canonical probabilities are independent,

$$\ln(p_{ij}^{\mathrm{A+B}}) = \ln(p_i^{\mathrm{A}}) + \ln(p_j^{\mathrm{B}}) \ \Rightarrow \ p_{ij}^{\mathrm{A+B}} = p_i^{\mathrm{A}} \cdot p_j^{\mathrm{B}} \ . \tag{11}$$



Equation (9) indicates that the result in Eq.(11) can be obtained simply by setting $\lambda_1=1$. Certainly, this restricts the generality, but it can be used as a trick to simplify the calculations. Furthermore, we can easily obtain the additivity of entropy. Indeed, applying the operator $\sum_{i=1}^{W}\sum_{j=1}^{W} p_{ij}^{A+B} \times$ on both sides of Eq.(11), we obtain

$$
\begin{aligned}
p_{ij}^{A+B} \ln p_{ij}^{A+B} &= p_{ij}^{A+B} \ln p_i^A + p_{ij}^{A+B} \ln p_j^B \\
\Rightarrow -\sum_{i=1}^{W}\sum_{j=1}^{W} p_{ij}^{A+B} \ln p_{ij}^{A+B} &= -\sum_{i=1}^{W} p_i^A \ln(p_i^A) - \sum_{j=1}^{W} p_j^B \ln(p_j^B)
\end{aligned}
\tag{12}
$$

5  because $\sum_{j=1}^{W} p_{ij}^{A+B} = p_i^A$ , $\sum_{i=1}^{W} p_{ij}^{A+B} = p_j^B$ . Hence, we arrive at the additivity of the entropy of the system to the entropies of the subsystems,

$$
S^{A+B} = S^A + S^B . \tag{13}
$$

### 3.3 Tsallis entropy

Next, we continue with the Tsallis $q$-entropy,

$$
S(p_1, p_2,...,p_W) = \frac{1 - \phi(p_1, p_2,...,p_W)}{q-1} = \frac{1}{q-1} \cdot \sum_{k=1}^{W} (p_k - p_k^{\,q}) . \tag{14}
$$

where the argument is defined by

$$
\phi(p_1, p_2,...,p_W) = \sum_{k=1}^{W} p_k^{\,q} . \tag{15}
$$

Again, the maximization of the entropy under the constraints of canonical ensemble involves maximizing the functional

$$
G(p_1, p_2,...,p_W) = \frac{1}{q-1} \cdot \sum_{k=1}^{W} \left(p_k - p_k^{\,q}\right) + \lambda_1 \sum_{k=1}^{W} p_k + \lambda_2 \sum_{k=1}^{W} p_k \varepsilon_k . \tag{16}
$$

15      Note that for simplicity we do not use the formulation of escort distributions (Beck and Schlogl 1993). The dyadic formalism of ordinary/escort distributions is of fundamental importance in the modern nonextensive statistical mechanics (Livadiotis 2017a; Chapter 1). It was shown that this dyadic formalism of distributions can be avoided in order to simplify the theory, but it leads to a dyadic formulation of entropy (Livadiotis 2017b).

Hence, $(\partial/\partial p_j)G(p_1, p_2,...,p_W)=0$ , gives

20

$$
p_j(\varepsilon_j) = \left[1 + (1-q^{-1}) \cdot (\lambda_1 - 1)\right]^{\frac{q^{-1}}{1-q^{-1}}} \cdot \left[1 + (1-q^{-1}) \cdot \frac{\lambda_2 \varepsilon_j}{1 + (1-q^{-1}) \cdot (\lambda_1 - 1)}\right]^{\frac{q^{-1}}{1-q^{-1}}} , \tag{17a}
$$

or





$$p_j(\varepsilon_j) = \exp_{q^{-1}}^{q^{-1}}(\lambda_1 - 1) \cdot \exp_{q^{-1}}^{q^{-1}}\left[\frac{\lambda_2 \varepsilon_j}{1_{q^{-1}}(\lambda_1 - 1)}\right] , \tag{17b}$$

where reflects a generalization of Eq.(4). We used the $Q$-deformed exponential function, and its inverse, the $Q$-logarithm function (Silva et al. 1998; Yamano 2002), defined by

$$\exp_Q(x) = [1 + (1-Q) \cdot x]_+^{-\frac{1}{Q-1}} , \quad \ln_Q(x) = \frac{1 - x^{1-Q}}{Q-1} . \tag{18a}$$

5    We also used the $Q$-deformed "unity function" (Livadiotis and McComas 2009), defined by

$$1_Q(x) = [1 + (1-Q) \cdot x]_+ , \tag{18b}$$

The subscript "+" in $[\ldots]_+$ denotes the cut-off condition, where $\exp_Q(x)$ becomes zero if its base $[\ldots]$ is non-positive. Therefore, Eq.(17b) leads to

$$p_j^{q-1} = 1 + (1 - q^{-1}) \cdot (\lambda_1 - 1) + (1 - q^{-1}) \cdot \lambda_2 \varepsilon_j , \tag{19}$$

$$\frac{1 - p_j^{q-1}}{q^{-1} - 1} = \ln_{q^{-1}}(p_j^q) = -q \cdot \ln_q(p_j^{-1}) = \lambda_2 \varepsilon_j + (\lambda_1 - 1) , \tag{20}$$

Dividing again the whole system in two subsystems A and B, using the additivity of energy, and setting $\lambda_1 = 1$, the independence relation (11) is generalized to

$$\ln_q[(p_{ij}^{A+B})^{-1}] = \ln_q[(p_i^A)^{-1}] + \ln_q[(p_j^B)^{-1}] \Rightarrow (p_{ij}^{A+B})^{q-1} = (p_i^A)^{q-1} + (p_j^B)^{q-1} - 1 , \tag{21}$$

which is sometimes called $q$-independence relation (Umarov et al. 2008). Then, we apply the operator $\sum_{i=1}^{W}\sum_{j=1}^{W} p_{ij}^{A+B} \times$,

$$\sum_{i=1}^{W}\sum_{j=1}^{W}(p_{ij}^{A+B})^q = \sum_{i=1}^{W}(p_i^A)^q + \sum_{j=1}^{W}(p_j^B)^q - 1 \Rightarrow \phi^{A+B} = \phi^A + \phi^B - 1 , \tag{22}$$

and using the entropic formula (14), we end up with the additivity of entropy, as shown in Eq.(13).

Note that the additivity leads to the extensivity: The additivity for some function $f$ is expressed by $f(A + B) = f(A) + f(B)$, or considering $N$ different subsystems,

$$f\left(\bigcup_{n=1}^{N} A_n\right) = \sum_{n=1}^{N} f(A_n) , \tag{23a}$$

20    while the extensivity is expressed by

$$f\left(\bigcup_{n=1}^{N} A_0\right) = N \cdot f(A_0) . \tag{23b}$$

Therefore, the canonical probability distribution, the one that maximizes the entropy under the constraints of canonical ensemble, makes the entropy additive (and therefore extensive) if the energy is additive. Several special conditions can simplify this result, e.g., constant Lagrange constraints with $\lambda_1 = 1$. This is true for both the entropic formulation of

25    classical BG and Tsallis nonextensive statistical mechanics.




Next, we will try to reverse the problem and seek to find the specific entropic formula, for which both the energy and entropy are additive.

## 4 Additive energy and entropy leads to Tsallis entropic formalism

The general entropic form is still function of the probabilities, $S = S\big(\{p_k\}_{k=1}^W\big)$. Then, its derivative with respect to any of the probability components, let's say the $i^{\text{th}}$, is also a function of all of these component, i.e., $\partial S / \partial p_i = F_i\big(\{p_k\}_{k=1}^W\big)$, for any $i$: $1,\dots,W$. However, the $2^{\text{nd}}$ constraint of the canonical ensemble connects the $i^{\text{th}}$ entropic derivative to some function $h_i$ of the $i^{\text{th}}$ energy, $\varepsilon_i$, namely, $\partial S / \partial p_i = h_i(\varepsilon_i)$. On the other hand, the canonical probability distribution derived from the entropy maximization constitutes an expression of the $i^{\text{th}}$ probability component with some function $g$ of the $i^{\text{th}}$ energy, $p_i = g(\varepsilon_i)$. Therefore, we conclude that $\partial S / \partial p_i = F_i(p_i)$, where $F_i = h_i \circ g^{-1}$; in other words, the entropy can be factorized as a summation of functions of each probability component, $S = \sum_{k=1}^W f_k(p_k)$, where we set $f_i(p_i) = \int F_i(p_i) dp_i$. Finally, we consider that none of the probability components should have special effect on the entropy, i.e., the whole distribution "weights" in the same way, so that the entropic functional $S = S\big(\{p_k\}_{k=1}^W\big)$ should be symmetric to any permutation of each components, e.g., $S = S(\dots, p_k, \dots, p_\ell, \dots) = S(\dots, p_\ell, \dots, p_k, \dots)$. This, leads to $f_k = f$; hence, considering (1) Entropy maximization, (2) No weighting, we obtain

$$S = \sum_{k=1}^W f(p_k) \ . \tag{24}$$

The maximization of entropy under the constraints of canonical ensemble, i.e., $1 = \sum_{k=1}^W p_k$ and $U = \sum_{k=1}^W p_k \varepsilon_k$, involves maximizing the functional $G\big(\{p_k\}_{k=1}^W\big) = \sum_{k=1}^W f(p_k) + \lambda_1 \sum_{k=1}^W p_k + \lambda_2 \sum_{k=1}^W p_k \varepsilon_k$. Hence, setting $\partial G\big(\{p_k\}_{k=1}^W\big) / \partial p_i = 0$, we obtain

$$F(p_i) + \lambda_1 + \lambda_2 \varepsilon_i = 0 \ , \text{ or } p_i(\varepsilon_i) = F^{-1}(-\lambda_1 - \lambda_2 \varepsilon_i) \ , \text{ with } F(x) \equiv f'(x) \ . \tag{25}$$

We now consider two systems A and B, with respective energy spectra $\{\varepsilon_i^A\}_{i=1}^W$ and $\{\varepsilon_j^B\}_{j=1}^W$, associated with the discrete probability distributions $\{p_i^A\}_{i=1}^W$ and $\{p_j^B\}_{j=1}^W$. The total system A+B has energy spectrum $\{\varepsilon_{ij}^{A+B}\}_{i,j=1}^W$, associated with the joint probability distribution $\{p_{ij}^{A+B}\}_{i,j=1}^W$. The probability distributions $\{p_i^A\}_{i=1}^W$ and $\{p_j^B\}_{j=1}^W$ are marginal of the joint distribution, i.e., $\sum_{j=1}^W p_{ij}^{A+B} = p_i^A$ and $\sum_{i=1}^W p_{ij}^{A+B} = p_j^B$. As we will find further below, the joint probability can be expressed as a function of the marginal probabilities, $p_{ij}^{A+B} = h(p_i^A, p_j^B)$. On the other hand, the relation between the joint





energies $\varepsilon_{ij}^{A+B}$ is rather trivial to be derived: particles in A with energy $\varepsilon_i^A$ and particles in B with energy $\varepsilon_j^B$ ensemble the

particles in A+B with energy $\varepsilon_{ij}^{A+B} = \varepsilon_i^A + \varepsilon_j^B$. Trivially, the same additivity holds for their mean values – the internal

energies,

$$U^{A+B} = \sum_{i,j} p_{ij}^{A+B} \varepsilon_{ij}^{A+B} = \sum_{i,j} p_{ij}^{A+B} \varepsilon_i^A + \sum_{i,j} p_{ij}^{A+B} \varepsilon_j^B = \sum_i p_i^A \varepsilon_i^A + \sum_j p_j^B \varepsilon_j^B = U^A + U^B \ . \tag{26}$$

5 Now, the probability distributions are related to their energies, according to Eq.(7). According to Eq.(25), we have

$$F\left(p_i^A\right) + \lambda_1 + \lambda_2 \varepsilon_i^A = 0 \ , \ F\left(p_j^B\right) + \lambda_1 + \lambda_2 \varepsilon_j^B = 0 \ , \ F\left(p_{ij}^{A+B}\right) + \lambda_1 + \lambda_2 \varepsilon_{ij}^{A+B} = 0 \ , \tag{27}$$

and due to the additivity of energies, we obtain

$$F\left(p_{ij}^{A+B}\right) - \lambda_1 = F\left(p_i^A\right) + F\left(p_j^B\right) . \tag{28}$$

Again, the Lagrange constant, $\lambda_1$ and $\lambda_2$, are considered to be constant. Setting $\tilde{F} \equiv \frac{1}{-\lambda_1} F$, Eq.(28) becomes

$$\left[\tilde{F}\left(p_{ij}^{A+B}\right) - 1\right] = \left[\tilde{F}\left(p_i^A\right) - 1\right] + \left[\tilde{F}\left(p_j^B\right) - 1\right] \ , \ or, \tag{29}$$

$$p_{ij}^{A+B} = h(p_i^A, p_j^B) , \text{ with } h(x,y) \equiv \tilde{F}^{-1}\left[\tilde{F}(x) + \tilde{F}(y) - 1\right] \ . \tag{30}$$

Then, we apply $\sum_{i=1}^W \sum_{j=1}^W p_{ij}^{A+B} \times$ in both sides of Eq.(29),

$$\sum_{i,j=1}^W \left[\tilde{F}\left(p_{ij}^{A+B}\right) - 1\right] p_{ij}^{A+B} = \sum_{i=1}^W \left[\tilde{F}\left(p_i^A\right) - 1\right] \sum_{j=1}^W p_{ij}^{A+B} + \sum_{j=1}^W \left[\tilde{F}\left(p_j^B\right) - 1\right] \sum_{i=1}^W p_{ij}^{A+B} \ , \ or$$

$$\sum_{i,j}^W \left[\tilde{F}\left(p_{ij}^{A+B}\right) - 1\right] p_{ij}^{A+B} = \sum_i^W \left[\tilde{F}\left(p_i^A\right) - 1\right] p_i^A + \sum_j^W \left[\tilde{F}\left(p_j^B\right) - 1\right] p_j^B \ . \tag{31}$$

15 (Note: The number of allowed states may be different for the two subsystems, WA≠WB, but here it does not make any

difference to consider $W_A=W_B=W$.)

We recall that $\tilde{F}(x) \equiv \frac{1}{-\lambda_1} f'(x)$, thus, we find

$$\sum_{i,j}^W \left[\frac{1}{-\lambda_1} f'\left(p_{ij}^{A+B}\right) - 1\right] p_{ij}^{A+B} = \sum_i^W \left[\frac{1}{-\lambda_1} f'\left(p_i^A\right) - 1\right] p_i^A + \sum_j^W \left[\frac{1}{-\lambda_1} f'\left(p_j^B\right) - 1\right] p_j^B \ . \tag{32}$$

We compare this property relation with the additivity of entropy

$$S^{A+B} = \sum_{i,j}^W f\left(p_{ij}^{A+B}\right) = \sum_i^W f\left(p_i^A\right) + \sum_j^W f\left(p_j^B\right) = S^A + S^B \ . \tag{33}$$

The two functions $f(x)$ and $[\frac{1}{-\lambda_1} f'(x) - 1] \cdot x$ have the same additivity property. Therefore, one function $f$ that can ensure for

the additivity of entropy is the one that obeys to the proportionality, $f(x) \propto [\frac{1}{-\lambda_1} f'(x) - 1] \cdot x$, or to the differential equation





$$f(x) = c \cdot \left[\frac{1}{-\lambda_1} f'(x) - 1\right] \cdot x \;,\; \text{or} \; f'(x) + \frac{\lambda_1}{c}\frac{1}{x} f(x) = -\lambda_1 \;, \tag{34}$$

with solution

$$f(x) = \lambda_1 \cdot \frac{x - x^{\frac{-\lambda_1}{c}}}{\frac{-\lambda_1}{c} - 1} + f(x=1) \cdot x^{\frac{-\lambda_1}{c}} \;. \tag{35}$$

(Note: The selection of proportionality between the two functions $f(x)$ and $\left[\frac{1}{-\lambda_1} f'(x) - 1\right] \cdot x$ makes the derivation of Eq.(34)

5  a sufficient but not necessary condition. Other functionals may also exist; for example, a linear combination of the two
mentioned functions.)

Then, we choose $f(x=1) = 0$, and we set $q \equiv \frac{-\lambda_1}{c}$, where find

$$f(x) = \lambda_1 \cdot \frac{x - x^q}{q - 1} \;, \tag{36}$$

or, setting also $\lambda_1 = 1$,

$$f(x) = \frac{x - x^q}{q - 1} \;. \tag{37}$$

Therefore, the entropic function $S = \sum_{k=1}^{W} f(p_k)$ becomes

$$S = \frac{1}{q-1} \cdot \sum_{k=1}^{W} \left( p_k - p_k{}^q \right) \;, \tag{38}$$

that is, the Tsallis entropic formulation that builds the nonextensive statistical mechanics.

## 5 Conclusions

15  The paper resolved a basic problem about the origin of the distributions and statistical mechanics applied in space plasmas.
Kappa distributions, or combinations thereof, can describe the velocities and energies of the plasma populations in space
plasmas. While these distributions were used since mid-60's for modeling space plasma datasets, their physical origin was
remaining unknown. It was just the last decade that was completely understood that the statistical origin of these
distributions is not the Boltzmann-Gibbs' classical statistical mechanics, but the Tsallis non-extensive statistical mechanics
20  (Livadiotis 2017a; Chapter 1). Indeed, the kappa distribution is the outcome of the maximization of the $q$-entropy of Tsallis
under the constraints of canonical ensemble. Once this concept was understood by the science community, the next question
was about the physical origin and reasoning of this entropic formula. This paper showed that the $q$-entropy, which is the
entropic formula that maximized leads to the kappa distribution, can be derived under simple first-principles and conditions,
namely, by considering that energy and entropy are both additive physical quantities.

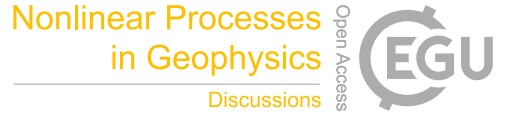

**Acknowledgements**: The work was supported in part by the project NNX17AB74G of NASA's HGI Program.

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
