# Peer review of "Derivation of the entropic formula for the statistical mechanics of space plasmas"

_Nonlinear Processes in Geophysics, 2017_

## Referee Comment (RC1) · Anonymous Referee #1 · 16 Oct 2017

This paper by George Livadiotis provides a derivation of the entropic formula for the statistical mechanics of space plasmas. The subject of the paper is very hot in view of the fact that the physical origin of the entropic formulation that leads to Kappa distributions, which describe velocities and energies of plasma populations in space plasmas, is still unknown. The present paper provides such a physical origin with a clear, original, concise and straightforward proof. I esteem that this proof by Livadiotis is very important and may become a cornerstone in this field. Since Nonlinear Processes in Geophysics (NPG) is an international and interdisciplinary journal for the publication of original research furthering knowledge on nonlinear processes in all branches of Earth, planetary, and solar system sciences, I also feel that the present paper obviously falls within the scope of NPG. For these reasons, I strongly recommend the publication of

this paper in NPG.

The treatment of the following minor points will strengthen the points made :

1)p.4, l.11 "(1960)"-> "(1957)"

2)p.4, l.22 "additivity)." -> "additivity."

3)p.9, l.6 "the 2nd constraint of" please clarify which constraint is considered second for the sake of the reader's convenience.

4)p.9, l.9 the existence of the inverse $(g^{-1})$ of the energy distribution function $g(\epsilon_i)$ is absolutely reasonable but it should it be mentioned for the sake of the reader's convenience.

5)p.10, l.9 "Again, the Lagrange constant, $\lambda_1$ and $\lambda_2$, are considered to be constant." please clarify.

6)p.10, Eq.(30) a symbol different from $h(x,y)$, e.g., $h_{A+B}(x,y)$, might be better to avoid reader's confusion with $h_i(\epsilon_i)$ of ll.6-7 of p.9.

7)p.10, l.15 "WA", "WB" -> "W$_A$", "W$_B$"

8)p.11, l.5 "functionals" -> "functional forms"

9)p.11, l.7 Please clarify that the selection of $f(x = 1) = 0$ is necessary from the condition $S[p_i = 1, p_j = 0 \ \forall \ j \neq i] = 0$.

―――――――――――――

---

## Referee Comment (RC2) · Anonymous Referee #2 · 9 Nov 2017

In this paper, the author present results which show that Tsallis entropy —whose maximum occurs when velocities follow the kappa distribution— can be derived for plasmas where entropy and energy are additive.

It is an interesting paper which deserves publication in this journal, but there are some issues which are not clear, and that need to be solved before accepting it. They are basically precisions on some statements or arguments that should be more explicit to follow the conclusions of the manuscript.

1. Page 1. Line 2. It is said that the fact that space plasmas follow kappa distributions is a "vastly different statistical behavior between classical systems and space plasmas". The sentence suggests that space plasmas are very special in

this sense, but they are rather just one example of a large family of systems where non-Maxwellian behavior is found. Kappa-like or Tsallis-like distribution functions can be found in spin systems, high-energy physics, turbulent fluids, etc., as well as many examples in biological or social systems. So this should be put in the proper context.

2. Page 1. Line 16. "The induction of any type of correlations. . . departs the system from thermal equilibrium to be re-stabilized to other stationary states. . . described by kappa distributions."

   This sentence is too strong. When correlations are absent, one should expect that the thermal equilibrium is Maxwellian. However, is there any guarantee, in general, that (a) any correlation leads to a non-Maxwellian equilibrium; (b) if it does, the final state is described by a kappa distribution?

   Besides, is it always correct to say that non-Maxwellian distributions mean absense of thermal equilibrium? Isn't it possible that a system reaches thermal equilibrium (in the sense that there is no further flow of energy between its components and with its surroundings) while having correlations?

3. Page 2. Line 34. The absense of collisions is not necessary to preserve correlations. On the contrary, they may well be the reason to preserve them.

4. Page 3. Line 1. Again, energy can be conserved in the presence of collisions.

5. Page 3. Line 1. In principle, energy can be conserved in systems where the energy cannot be separated as the sum of individual particle energies. Please explain.

6. The main claim in the paper seems to be supported by the final paragraphs in Sec. 2 (pages 4 and 5). Although the Tsallis entropy is in general non-additive, there are certain correlations, depending on a certain function $g(x)$, which make

it additive again. And then an expression for the function $g$ is given, which leads to the Tsallis entropy.

However, this leaves the impression that this is a particular case, which allows to recover the Tsallis entropy, but does not help to understand why the Tsallis entropy should be the "right" form to describe correlated systems.

Can all physically acceptable correlations be expressed as $p_{ij}^{A+B} = g^{-1}[g(p_i^A) + g(p_j^B)]$? And is there any argument why one would expect that this holds specifically for plasmas? (Or space plasmas, which is the subject of this paper.)

One could make an *a posteriori* argument, since Tsallis entropy is known to lead to kappa distributions naturally, but the paper seems to make more general claims.

Given the above, some of the sentences in the conclusions are not clear. It says that "The paper resolved a basic problem about the origin of the distributions... in space plasmas", and that the $q$-entropy can be derived by considering additive energy and entropy. However, this seems to be true as long as the assumptions on the correlations [Eq. (21)] are correct, and there is no argument on its validity either for general systems or for plasmas in particular. Thus, it is not clear if the paper resolves a basic problem.

Please make more explicit arguments for these statements.

7. Page 9. Line 11. Entropy is stated to be symmetric on probabilities, arguing that "none of the probability components should have special effect on the entropy". This is too vague and should be rephrased. In the canonical formalism some states are more probable that others. What does "special effect" or "equal weights" means, then? Maybe it is an argument on the states rather than on the probabilities: relabeling the states does not change the entropy?

8. Tsallis entropy was proposed in the late 80s, and it was always proposed as a way to model the ubiquity of power-law/kappa distributions, understanding that they

are "equilibrium" (maximum entropy) configurations, but for an entropy different to the Boltzmann's one.

It is thus not clear why the paper says in the conclusions that "it was just in the last decade that was completely understood that the statistical origin of these distributions is not the Boltzmann-Gibbs' classical statistical mechanics."

Please explain.

A few additional formal issues:

1. Page 5, line 4: "both the formalisms".

2. Page 9, line 5: "these component"

3. Page 10, line 15: "$WA \neq WB$".

---

## Referee Comment (RC3) · Anonymous Referee #3 · 13 Nov 2017

This is a review article and review articles most of times are like a binary variable, either 0 or 1. This paper is definitely "1" and deserves publication.

Kappa distributions have been used in Space plasma physics for more than half a century and even longer in other fields. It is well-known in literature that kappa distributions characterizing space plasmas can be linked to the so-called Tsallis entropy, a non-extensive entropic form that generalizes the classical Boltzmann-Gibbs entropy. The Tsallis entropy after maximization leads to the exact form of kappa distribution. However, this does not consist an actual proof or a justification for the use of kappa distribution, as the "natural" question then is what is the origin of this type of entropy.

This paper has a single, yet import cause, and that is to reply actually to the physical origin of the Tsallis entropy and thus the origin of Kappa distributions. This is achieved

by showing that Eq. (36) emerges by two assumptions, i.e., the extensivity of energy leading to Eq. (32), and the extensivity of entropy in Eq. (33). The combination of these equations leads to the differential equation given in Eq. (34), which is essentially the solution of Eq. (36).

I believe that this is very crucial, as it is the first time that a "rational" origin is given to the entropic form characterizing the particles in space plasmas that are described by kappa distributions. This single important result justifies the publication of the paper.

Finally, regarding the Tsallis entropy, I would like to comment that as a form it was introduced by Havrda & Charvát (1967) under the name "structural a-entropy". I believe it is nice to mention that. Also, it is common in other fields, (information theory) the BG entropy to be named as Shannon entropy.

The rest of my comments are just typographical errors: (1) in "q-entropy" the "q" is sometimes in italics and sometimes not; (2) page 3/line 16: the BG acronym is given later in page 4; (3) sometimes the dash symbol is used instead of the minus symbol, e.g., page 6/line 18; (4) add spaces before and after "=", e.g., page 7/line1; (5) in equation (35) maybe it is better to use f(1) instead of f(x=1) if I am not missing something.

Refs. Havrda, J., & Charvát, F. (1967). Concept of structural a-entropy. Kybernetika, 3, 30–35.
* * *

---

## Author Comment (AC1) · 16 Nov 2017

MS No.: npg-2017-54 Title: "Derivation of the entropic formula for the statistical mechanics of space plasmas" Author: George Livadiotis

Respond to Referee #1

Thank you for your valuable comments that have improved the paper significantly. Thank you also for your good words, especially for finding the paper "very hot" and that it is very important and may become a cornerstone in this field. The revised version has been prepared by taking care all of your comments and corrections.

George Livadiotis

Please also note the supplement to this comment:
https://www.nonlin-processes-geophys-discuss.net/npg-2017-54/npg-2017-54-AC1-supplement.pdf
* * *
[Figure]

**Supplement:**

MS No.: npg-2017-54
Title: "Derivation of the entropic formula for the statistical mechanics of space plasmas"
Author: George Livadiotis

Respond to Referee #1

Thank you for your valuable comments that have improved the paper significantly. Thank you also for your good words, especially for finding the paper "very hot" and that it is very important and may become a cornerstone in this field. The revised version has been prepared by taking care all of your comments and corrections.

Together with this response, we have included a "track changes" version as well as a clean version of the revised manuscript. In the "track changes" version we mark all the changes-corrections in regards to the comments of all referees.

George Livadiotis

---

## Author Comment (AC2) · 16 Nov 2017

MS No.: npg-2017-54 Title: "Derivation of the entropic formula for the statistical mechanics of space plasmas" Author: George Livadiotis

Respond to Referee #2

Thank you for your valuable comments that have improved the paper significantly. Thank you also for finding the paper interesting and that deserves publication. The revised version has been prepared by taking care all of your comments and resolving the confusing issues you have mentioned. Together with this response, we have included a "track changes" version as well as a clean version of the revised manuscript. In the "track changes" version we mark all the changes-corrections in regards to your

comments. Below we reply to each of your comments separately.

1. Page 1. Line 2. It is said that the fact that space plasmas follow kappa distributions is a "vastly different statistical behavior between classical systems and space plasmas". The sentence suggests that space plasmas are very special in this sense, but they are rather just one example of a large family of systems where non-Maxwellian behavior is found. Kappa-like or Tsallis-like distribution functions can be found in spin systems, high-energy physics, turbulent fluids, etc., as well as many examples in biological or social systems. So this should be put in the proper context.

The referee is right. The text has been revised accordingly. Thanks!

2. Page 1. Line 16. "The induction of any type of correlations. . . departs the system from thermal equilibrium to be re-stabilized to other stationary states. . . described by kappa distributions." This sentence is too strong. When correlations are absent, one should expect that the thermal equilibrium is Maxwellian. However, is there any guarantee, in general, that (a) any correlation leads to a non-Maxwellian equilibrium;

Yes. This has been proved in Livadiotis & McComas 2011b, and generalized further in Livadiotis 2015c and 2017a, Chapter 5.

(b) if it does, the final state is described by a kappa distribution?

Yes! The existence of particle stationary states characterized by both (i) temperature, and (ii) correlations, means necessarily the formation of kappa distributions (or, combinations thereof). Particle systems, with or without correlations, may exist in other formulations, but they will not be characterized by a physically meaningful definition of temperature as follows from thermodynamic laws.

Besides, is it always correct to say that non-Maxwellian distributions mean absence of thermal equilibrium?

No. If the system is thermodynamically stabilized into a stationary state, then any non-Maxwellian distribution means: the distribution can be written as a combination or

superposition of kappa distributions, the existence of correlations, and the system is not at thermal equilibrium.

Isn't it possible that a system reaches thermal equilibrium (in the sense that there is no further flow of energy between its components and with its surroundings) while having correlations?

No. That is reaching a stationary state. The classic understanding of thermal equilibrium is a stationary state, but not all stationary states are in thermal equilibrium. The special about it is no correlations and Maxwellian behavior. All the $\kappa$- (or q-) stationary states could have similar characteristics with thermal equilibrium but with the highlighting property of local correlations among the particles.

3. Page 2. Line 34. The absense of collisions is not necessary to preserve correlations. On the contrary, they may well be the reason to preserve them.

Thermal collisions destroy correlations. Organized collisions may not.

4. Page 3. Line 1. Again, energy can be conserved in the presence of collisions.

In the presence of collisions, energy may be conserved or not. But in the absence of collisions it can be certainly be conserved, unless other particle interactions are taken place.

5. Page 3. Line 1. In principle, energy can be conserved in systems where the energy cannot be separated as the sum of individual particle energies. Please explain.

You are right! The text has been revised. Thank you!

6. The main claim in the paper seems to be supported by the final paragraphs in Sec. 2 (pages 4 and 5). Although the Tsallis entropy is in general non-additive, there are certain correlations, depending on a certain function g(x), which make it additive again. And then an expression for the function g is given, which leads to the Tsallis entropy. However, this leaves the impression that this is a particular case, which allows

to recover the Tsallis entropy, but does not help to understand why the Tsallis entropy should be the "right" form to describe correlated systems. Can all physically acceptable correlations be expressed as pA+Bij = g−1[g(pAi) +g(pBj)]? And is there any argument why one would expect that this holds specifically for plasmas? (Or space plasmas, which is the subject of this paper.) One could make an a posteriori argument, since Tsallis entropy is known to lead to kappa distributions naturally, but the paper seems to make more general claims. Given the above, some of the sentences in the conclusions are not clear. It says that "The paper resolved a basic problem about the origin of the distributions. . . in space plasmas", and that the q-entropy can be derived by considering additive energy and entropy. However, this seems to be true as long as the assumptions on the correlations [Eq. (21)] are correct, and there is no argument on its validity either for general systems or for plasmas in particular. Thus, it is not clear if the paper resolves a basic problem. Please make more explicit arguments for these statements.

- The BG entropy and the produced Maxwell-Boltzmann distribution follow a certain correlation type, that is, no correlation, as given by Eq.(11): . Section 2 is based on one of the following three equivalent assumptions: (i) BG entropic formulation; (ii) Maxwell-Boltzmann distributions in the canonical ensemble; (iii) no correlation or independence, as given by Eq.(11). - The Tsallis entropy and the produced (canonical ensemble) q-exponential or kappa distributions follow a certain correlation type, that is the one given in Eq.(21): . Section 3 is based on one of the following three equivalent assumptions in the canonical ensemble: (i) Tsallis entropic formulation; (ii) q-exponential or kappa distributions; (iii) Special Correlation type called also q-independence, as given by Eq.(21). - There is no assumption on correlations to produce the main claim of the paper. The only assumptions are the additivity of the entropy and energy. Section 4 is based on these assumptions to produce that Tsallis entropy describes particle systems such as space plasmas; thus, kappa distributions too; and thus, the correlations of Eq.(21), too. In addition, superposition of stationary states can be described by more complicated distributions or correlations (e.g., see Livadiotis & McComas 2013a;

Livadiotis 2017a, Chapter 4).

7. Page 9. Line 11. Entropy is stated to be symmetric on probabilities, arguing that "none of the probability components should have special effect on the entropy". This is too vague and should be rephrased. In the canonical formalism some states are more probable that others. What does "special effect" or "equal weights" means, then? Maybe it is an argument on the states rather than on the probabilities: relabeling the states does not change the entropy?

Correct, it is an argument on the states, and the text has now been rephrased. Thanks!

8. Tsallis entropy was proposed in the late 80s, and it was always proposed as a way to model the ubiquity of power-law/kappa distributions, understanding that they are "equilibrium" (maximum entropy) configurations, but for an entropy different to the Boltzmann's one. It is thus not clear why the paper says in the conclusions that "it was just in the last decade that was completely understood that the statistical origin of these distributions is not the Boltzmann-Gibbs' classical statistical mechanics." Please explain.

Empirical Kappa distributions have been used without any statistical framework for almost half a century. The connection of these distributions with the statistical framework of non-extensive statistical mechanics is the one that has been completed about a decade ago. Corrections have been made to resolve the confusion. Thanks!

A few additional formal issues: 1. Page 5, line 4: "both the formalisms". 2. Page 9, line 5: "these component" 3. Page 10, line 15: "WA 6= WB".

All corrected. Thanks!

Again, thank you for your valuable comments and the effort to improve this work.

George Livadiotis

Please also note the supplement to this comment:

https://www.nonlin-processes-geophys-discuss.net/npg-2017-54/npg-2017-54-AC2-supplement.pdf

---

## Author Comment (AC3) · 16 Nov 2017

MS No.: npg-2017-54 Title: "Derivation of the entropic formula for the statistical mechanics of space plasmas" Author: George Livadiotis

Respond to Referee #3

Thank you for your valuable comments that have improved this work. Thank you also for finding that definitely deserves publication. The revised version has been prepared by taking care all of your comments and corrections.

George Livadiotis

Please also note the supplement to this comment:

[Figure]

https://www.nonlin-processes-geophys-discuss.net/npg-2017-54/npg-2017-54-AC3-supplement.pdf

**Supplement:**

MS No.: npg-2017-54
Title: "Derivation of the entropic formula for the statistical mechanics of space plasmas"
Author: George Livadiotis

Respond to Referee #3

Thank you for your valuable comments that have improved this work. Thank you also for finding that definitely deserves publication. The revised version has been prepared by taking care all of your comments and corrections.

Together with this response, we have included a "track changes" version as well as a clean version of the revised manuscript. In the "track changes" version we mark all the changes-corrections in regards to all referees' comments.

George Livadiotis

---

## Author Comment (AC6) · 22 Nov 2017

MS No.: npg-2017-54 Title: "Derivation of the entropic formula for the statistical mechanics of space plasmas" Author: George Livadiotis

Respond to Referee #3

Thank you for your valuable comments that have improved this work. Thank you also for finding that definitely deserves publication. The revised version has been prepared by taking care all of your comments and corrections.

George Livadiotis

Please also note the supplement to this comment:

[Figure]

https://www.nonlin-processes-geophys-discuss.net/npg-2017-54/npg-2017-54-AC6-supplement.pdf
* * *
Interactive
comment

[Figure]

**Supplement:**

MS No.: npg-2017-54
Title: "Derivation of the entropic formula for the statistical mechanics of space plasmas"
Author: George Livadiotis

Respond to Referee #3

Thank you for your valuable comments that have improved this work. Thank you also for finding that definitely deserves publication. The revised version has been prepared by taking care all of your comments and corrections.

Together with this response, we have included a "track changes" version as well as a clean version of the revised manuscript. In the "track changes" version we mark all the changes-corrections in regards to all referees' comments.

George Livadiotis

---

## Referee Report (RR1)

The revised version of the manuscript has considered some of my comments to my previous version, and has clarified some of the issues in the response.

However, there are a few issues which need to be improved in order to provide better arguments for the paper's results.

2) Reading references Livadiotis and McComas 2011b, and Livadiotis 2015c, I do not find that they specifically demonstrate that any correlation leads to a non-Maxwellian equilibrium. I understand that they show how correlations are related to the $\kappa$ index, but starting from a velocity distribution already given by a kappa function.

   But if a system has correlations, of any kind, and no assumption on the underlying distribution is made, will this inevitably lead to a kappa function? The references given do not seem to address this.

   I note that the author mentions at some point "kappa distributions or combinations thereof". Which is a delicate argument in my opinion, because if an arbitrary distribution can be regarded as a combination of kappa distributions, then one should ask whether it would be mathematically possible to consider it as a combination of some other family of distributions. In which case the kappa distribution would not have a special status.

   Anyway, if the author can clarify this and provide adequate references, this should be included in the manuscript.

3) The argument in the second paragraph of page 3 is not very clear, regarding the role of collisions in the persistence of correlations and of Tsallis distributions.

   Since kappa/Tsallis distribution functions can be found in a variety of systems, both collisionless and collisional, then it should be more clear that the lack of collisionality could work as an argument for the particular case of space plasmas. Or maybe the author is, in fact, discussing this, by finding more general reasons for the occurrence of kappa distributions? If so, this emphasis should be more clear in the manuscript. If not, then it should be noted that it is one possible route for preserving correlations, which works for space plasmas.

5) In the new text, it is mentioned that weakly coupled plasmas can be described as ideal gases, from which the additivity of energy follows. However, in ideal gases particles are non-interacting, and thus should be Maxwellian. Is "ideal gas" the correct expression here?

---

## Author Response (AR2)

MS No.: npg-2017-54 – Second Review

Title: "Derivation of the entropic formula for the statistical mechanics of space plasmas"

Author: George Livadiotis

Respond to Referee #2

    The author is grateful for these new comments that have all taken into account and improved the paper. Below there is a reply to each of the three comments separately.

    Together with this response, we have included a "track changes" version of the revised manuscript. In the "track changes" version we mark all the changes-corrections in regards to your comments:

George Livadiotis

Reply to each comment:

A) Reading references Livadiotis and McComas 2011b, and Livadiotis 2015c, I do not find that they specifically demonstrate that any correlation leads to a non-Maxwellian equilibrium. I understand that they show how correlations are related to the κ index, but starting from a velocity distribution already given by a kappa function. But if a system has correlations, of any kind, and no assumption on the underlying distribution is made, will this inevitably lead to a kappa function? The references given do not seem to address this.

I note that the author mentions at some point "kappa distributions or combinations thereof". Which is a delicate argument in my opinion, because if an arbitrary distribution can be regarded as a combination of kappa distributions, then one should ask whether it would be mathematically possible to consider it as a combination of some other family of distributions. In which case the kappa distribution would not have a special status. Anyway, if the author can clarify this and provide adequate references, this should be included in the manuscript.

    No, kappa distributions are interwoven with a certain type of correlations. This is precisely mentioned in page 5:

<< Nevertheless, certain correlations, expressed by the relation $p_{ij}^{A+B} = g^{-1}[g(p_i^A) + g(p_j^B)]$ for some function $g$, can make the Tsallis entropy additive. … The probability distribution that maximizes the Tsallis entropy under the same constraints obeys to specific correlations expressed by $g(x) \propto (x^{q-1} - 1)/(q-1)$ or $(p_{ij}^{A+B})^{q-1} = (p_i^A)^{q-1} + (p_j^B)^{q-1} - 1$, which makes again the entropy additive.>>

Nevertheless, in the revised version, we have added a clarifying note and references that while single kappa distributions induce a certain type of correlation, that is, a certain relationship among the probabilities, this correlation can be further generalized when a combination or superposition of kappa distributions is taken into account; e.g., see: Spectral Statistics, Tsallis 2009, Linear/Nonlinear superposition, Chapter 6.2.1; Livadiotis and McComas 2013a, Appendix A; Livadiotis 2017a, Chapter 4.3.4.

B) The argument in the second paragraph of page 3 is not very clear, regarding the role of collisions in the persistence of correlations and of Tsallis distributions. Since kappa/Tsallis distribution functions can be found in a variety of systems, both collisionless and collisional, then it should be more clear that the lack of collisionality could work as an argument for the particular case of space plasmas. Or maybe the author is, in fact, discussing this, by finding more general reasons for the occurrence of kappa distributions? If so, this emphasis should be more clear in the manuscript. If not, then it should be noted that it is one possible route for preserving correlations, which works for space plasmas.

You are right! Indeed, our claim works for particle systems such as space plasmas, where the collisions can destroy correlations, and thus, their collective behavior. We have made this clarification. Thanks!

C) In the new text, it is mentioned that weakly coupled plasmas can be described as ideal gases, from which the additivity of energy follows. However, in ideal gases particles are non-interacting, and thus should be Maxwellian. Is "ideal gas" the correct expression here?

It is a standard characterization of space plasmas noted in textbooks that space plasmas are described as ideal gases; also, they are characterized by correlations. The key of both characterizations is the week long-range interactions:

[revised manuscript text omitted]